# Aging Analysis of Thermally Aged Asphalt Using Peak-Fitting Method: Its Pattern and Statistical Prediction

Zhuolin Li [1,*], Shuolei Huang [1], Kunyang Zhao [2,3], Zonghe Li [2,3] and Panfei Zheng [2,3]

1   Liaoning Transportation Research Institute Co., Ltd., Key Laboratory of Transport Industry of Expressway Maintenance Technology, Shenyang 110015, China; h_shuolei@126.com
2   Beijing Super-Creative Technology Co., Ltd., Beijing 100621, China; zhaoky@cacc.com.cn (K.Z.); lizonghe@cacc.com.cn (Z.L.); zhengpf@cacc.com.cn (P.Z.)
3   China Airport Construction Group Co., Ltd., Beijing 100101, China
*   Correspondence: 15771955902@163.com; Tel.: +86-150-40026702

**Abstract:** In this study, the peak fitting method was proposed to deal with and analyze the thermally aged pattern and statistical prediction of asphalt in order to reduce the calculation accuracy caused by overlapping and partial overlapping of infrared spectrum peaks. The aromatic functional group index, aliphatic branch chain index, aliphatic functional group index, butadiene functional group index, styrene index and carbonyl functional group index were used to evaluate the aged asphalt. The piecewise fitting method of OriginPro 9.0 was proposed to investigate the thermally aged pattern of asphalt based on the peak fitting method. From the testing results, after the process of thermal aging, the aromatic functional group index and carbonyl functional group index increased, the aliphatic branch chain index, aliphatic functional group index and butadiene functional group index decreased, and the styrene index was stable. Taking the carbonyl functional group index as the research object, the thermally aged pattern of asphalt based on the peak fitting method conformed to the two-reaction kinetic model and the aging process consists of a fast-rate reaction period and constant-rate reaction period, where the coefficients of determination are all above 0.91. These results show that the pattern and statistical prediction of thermally aged asphalt by infrared spectrum illustrate rapidly for a time and then develop at a relatively constant speed, which corresponds to the road performance of asphalt aging. Therefore, the aging condition of asphalt can be evaluated, which has important theoretical significance to predict the aging state of asphalt and accurately grasping the road aging state.

**Keywords:** asphalt; aging; carbonyl; infrared spectrum; peak fitting method

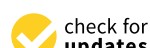



## 1. Introduction

The aging mechanism of asphalt binder is complex, while aging of the binder has significant impacts on the long-term performance and service life of pavement [1–8]. With the increase in the service time, the asphalt binder gradually becomes stiffer and more brittle, and the decay of asphalt performance will cause the pyrolysis of asphalt and fatigue damage of asphalt pavement [9–13]. In the process of asphalt aging, oxygen and time play important roles in promoting the aging degree of asphalt as two important aging factors. From the perspective of chemical reactions in oxidative aging, the oxygen reacts with the activity group and produces the polar molecules in the asphalt binder, such as carbonyl compounds and sulfoxide group, and with the increase in polar molecules, the aging degree of asphalt gradually becomes serious [14–17], which causes the performance of asphalt to decay. Therefore, there is a crucial demand to study the thermally aged pattern of asphalt for estimating the state of aging asphalt and predicting the performance of asphalt binder.

The thermal oxidative aging pattern of asphalt has been attracting increasing attention in recent years. Research shows that the aging process follows the dual sequential aging mechanism, and the aging degree of asphalt can be characterized by carbonyl content. Moreover, aged binder under different aging temperatures and durations can be measured

by an FTIR (FourierTransform Infrared) spectrometer. It is assumed that the growth rate of carbonyl content in the aging process can be described by the Arrhenius kinetics theory [18–24]. Additionally, the oxidative aging process can occur in two parallel steps; one is a nonlinear rapid reaction period, and the other is a constant and slower reaction period [18–20]. Much effort has been devoted to thermally aged asphalt. Liu et al. [21] demonstrated that the carbonyl area between $1650\,\mathrm{cm}^{-1}$ and $1820\,\mathrm{cm}^{-1}$ was linearly related to the oxygen content that reacted with asphalt. Jin et al. [22] proposed the importance of the fast-rate reaction period and found the Fast-Rate and Constant-Rate Oxidation Kinetics Model could predict the change in carbonyl content in the thermally aged process, and this model was subsequently called the two-reaction kinetic model. Huang et al. [23] also created a two-reaction kinetic model based on the peak height of the carbonyl and sulfoxide group. In conclusion, it is feasible and accurate to predict the carbonyl content of aged asphalt binder using a two-reaction kinetic model, and the two-reaction kinetic model under conditions of thermal aging can be used to predict the aging degree of asphalt. However, previous studies on the prediction of asphalt aging degree use the calculation results of the infrared spectrum directly, without any handling, overlooking the fact that the characteristic peaks of each group are not all individual, and the group peaks often overlap or partially overlap, which will deeply affect the accuracy of the calculation results.

In this work, the thermally aged pattern of asphalt based on the peak fitting method was investigated using statistical methods. Fourier transform infrared spectroscopy combined with the mathematical method of peak fitting were conducted to investigate the changing trend of asphalt group subjected to oxidative aging. Changes in carbonyl area (CA) under multiple durations provided kinetics parameters to be used in an asphalt pavement oxidation.

## 2. Materials and Methods

### 2.1. Materials

Two distinct binders were considered in this study: base bitumen (Liaohe 90#) and SBS-modified bitumen (polymer occupies 5%). The performance tests were carried out according to "Test specification for asphalt and asphalt mixture of Highway Engineering", JTG E20-2011, and the two kinds of bitumen met the requirements of the specification, as shown in Tables 1 and 2.

**Table 1.** Liaohe 90# base bitumen index.

| Index | Test Value | Technical Standard | Test Method |
| --- | --- | --- | --- |
| 25 °C Penetration/0.1 mm | 84 | 80–100 | T0604 |
| 15 °C Ductility/cm | ≥100 | ≥100 | T0605 |
| Softening point/°C | 46.5 | ≥45 | T0606 |

**Table 2.** SBS-modified bitumen index.

| Index | Test Value | Technical Standard | Test Method |
| --- | --- | --- | --- |
| 25 °C Penetration/0.1 mm | 52 | ≥50 | T0604 |
| 15 °C Ductility/cm | 50 | ≥45 | T0605 |
| Softening point/°C | 80.0 | ≥70 | T0606 |

### 2.2. Aging Procedure

The aging procedure was carried out based on the rolling thin film oven aging test [9], in which the two different asphalts were poured into aging bottles with $35\,\mathrm{g} \pm 0.5\,\mathrm{g}$ weight, respectively. After preparation of asphalt samples, they were aged under a standard temperature of $163 \pm 0.5\,°C$ and air flow rate under $4000 \pm 200\,\mathrm{mL/min}$ in accordance with different aging durations, and the samples were collected for testing according to the schedule in Table 3.

**Table 3.** Sampling timeline for thermally aged asphalt.

| Aging Temperature (K) | Air Pressure | Aging Time (h) |
|:---:|:---:|:---:|
| $436.15 \pm 0.5$ | $4000 \pm 200$ mL/min | 1.3, 3, 5, 7, 9, 12, 14, 16, 19, 21, 24, 26, 28, 30, 32, 34, 36.5, 38, 40, 42, 44, 46, 48 |

### 2.3. Asphalt Group Measurement

The asphalt groups at different aging durations were measured using an Agilent hand-held infrared spectrometer 4300 TopScan FTIR (Agilent technology, Penang, Malaysia), with a resolution of 4 cm$^{-1}$, a scan number of 32 and a wave number between 4000 and 650 cm$^{-1}$, and the sample spectra were obtained by the ATR accessory of an infrared spectrometer directly. In this study, in order to improve the data accuracy and reduce the data dispersion, the infrared spectrum of each aging asphalt sample was sampled 15 times, and the average value of the group absorbance area was taken as the sample data, in which the standard deviation was less than 3%.

### 2.4. Peak Fitting Method

Peak fitting is an effective method that adopts some algorithms in order to separate the overlapping peaks of FTIR and calculate the group content quantitatively. Peakfit software (v4.12) was used in this paper, and after the baseline correction and Savitsky Golay smoothing, the spectrum was fitted by the Gauss fitting function. Finally, according to the results of the experiments, R2 was required to be above 0.90, and iteration was about 7. The fitting method is shown in Figure 1.

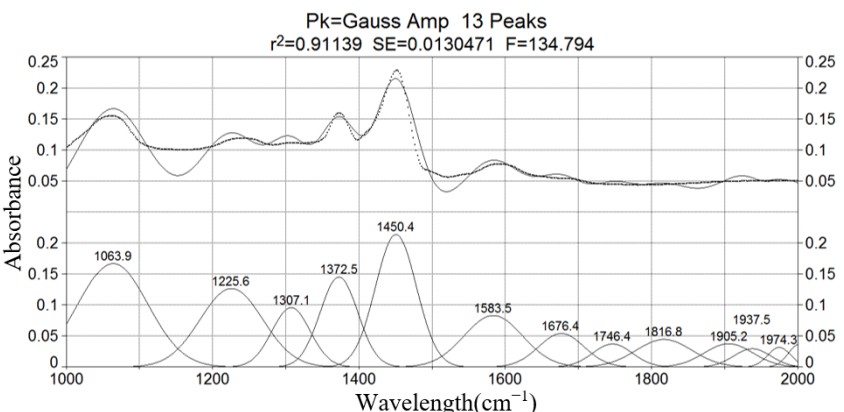

**Figure 1.** Peak fitting method.

## 3. Results and Discussion

### 3.1. Infrared Spectrum Analysis of Original Materials

Infrared spectrum scanning was carried out on the unaged original asphalt and SBS-modified asphalt to obtain the spectrum shown in Figure 2.

The chemical composition of the base asphalt was mainly composed of aliphatic compounds, aromatic compounds and heteroatom derivatives. The peaks at 2920 cm$^{-1}$ and 2850 cm$^{-1}$ were caused by the antisymmetric stretching vibration and symmetric stretching vibration of saturated hydrocarbon $CH_2$, the peak at 1600 cm$^{-1}$ was caused by respiratory vibration of asymmetric substituted benzene ring in asphalt, and the peak at 1450 cm$^{-1}$ was caused by shear vibration of methylene ($-CH_2-$), while the peak at 1375 cm$^{-1}$ was caused by umbrella vibration of methyl ($-CH_3$).

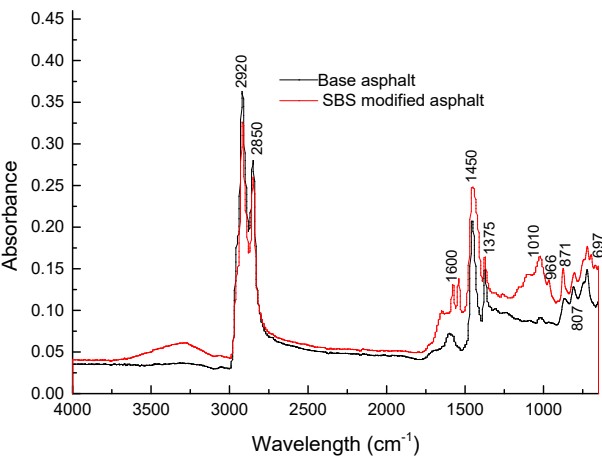

**Figure 2.** Infrared spectrum of base asphalt and SBS-modified asphalt.

Compared to the base of the asphalt, besides aliphatic compounds, aromatic compounds and heteroatom derivatives composition, the SBS-modified asphalt also contained a styrene butadiene styrene component in the SBS modifier [25–27]. The small absorption peak at about 697 cm$^{-1}$ was caused by the C–H deformation vibration of a single substitute of the polystyrene benzene ring, which proved the existence of SBS. The weak absorption peak at nearly 749 cm$^{-1}$ was caused by the formal out-of-plane rocking vibration peak of 1,4-polybutadiene, while the strong absorption peaks at 908 and 967 cm$^{-1}$ were originated by the bending vibration peak of polyolefin $RCH = CH_2$. A detailed analysis of characteristic absorption peaks is shown in Table 4 [28–35].

**Table 4.** Detailed analysis of characteristic absorption peaks.

| Wave Numbers/cm$^{-1}$ | Attribution | Corresponding Composition |
|---|---|---|
| 3419 | (–OH) Symmetric and antisymmetric stretching vibration | Liquid water |
| 2922/2852 | Methy(–CH$_2$–) symmetry vibration | Aliphatic long chain (saturate) |
| 1680–1820 | Carbonyl | C=O vibration |
| 1600 | Asymmetric benzene ring breathing vibration | Benzene ring and carboxyl |
| 1461 | Methylene(–CH$_2$–) shear type vibration | Aliphatic long chain (saturate) |
| 1427 | Anti-symmetric stretching vibration | Calcite (carbonate compounds) |
| 1377 | Methyl (–CH$_3$) umbrella vibration | Aliphatic branched chain (saturate) |
| 1163 | (SO$_2$) Symmetric stretching | Aliphatic sulfonic acid |
| 1080 | (CCL) Vibration | Aromatic stretching |
| 1032 | Sulfoxide (S=O) stretching vibration | Oxidation of sulfur |
| 966 | Butadiene stretching vibration | Butadiene(SBS) |
| 876 | Out-of-plane bending vibration | Calcite (carbonate compounds) |
| 712 | In-plane bending vibration | Calcite (carbonate compounds) |
| 698 | C–H deformation vibration of monosubstituents of benzene ring | Polystyrene benzene ring |

### 3.2. Infrared Spectrum Analysis of Aged Asphalt

The infrared spectra of the base asphalt and SBS-modified asphalt were measured after thermal aging for different times; following the smoothing process, the spectra were obtained as shown in Figure 3.

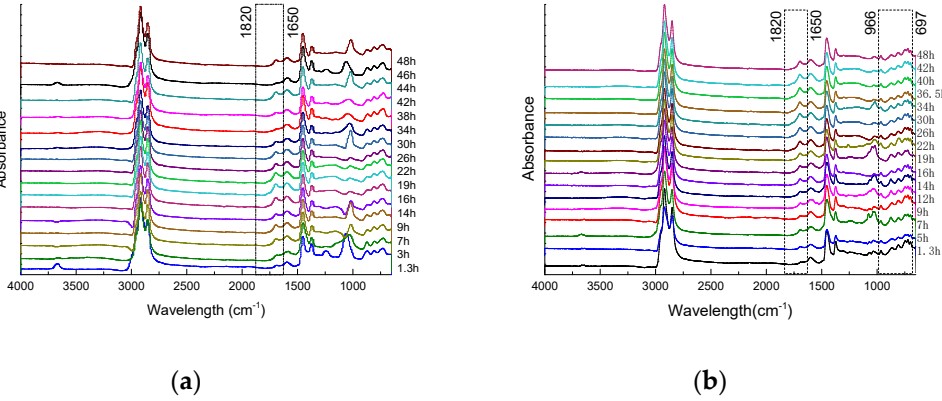

**Figure 3.** Infrared spectrum of base asphalt and SBS-modified asphalt after thermal aging: (**a**) base asphalt; (**b**) SBS-modified asphalt.

To quantitatively analyze the functional group content of asphalt under different thermal aging conditions and study the change trend of the functional groups, the Lambert Beer law was used, which indicated that when a beam of parallel monochromatic light passes vertically through an absorbing material, its absorbance is directly proportional to the concentration of the material; therefore, the peak area can be used to evaluate the aging condition. From the curve fitting results, in order to reduce variability, the ratio of the peak area of wave number 2922, 2852, 1600, 1450, 1370, 966, 698 cm$^{-1}$ to area $\Sigma A2000$ cm$^{-1}$ − 600 cm$^{-1}$ or $\Sigma A1370$ cm$^{-1}$ + 1460 cm$^{-1}$ were used to characterize the aging state of the asphalt. Therefore, the infrared spectrum characteristic peak index of the aromatic functional group index, the fat branched chain index, the aliphatic functional group index, the butadiene functional group index, the styrene index and the carbonyl functional group index were proposed, as shown in Equations (1)–(6). The aromatic functional group index $I_{Ar}$ represents changes in the aromatic functional groups (aromatics, resins, asphaltenes, etc.), and the aliphatic branched chain index $I_{B,a}$ represents the content of aromatic components in asphalt. The aliphatic functional group index $I_B$ was used to characterize the content of saturated components in asphalt. The butadiene index $I_{C=C}$ and styrene index $I_{C-H}$ were used characterize the content of polybutadiene and polystyrene in SBS-modified asphalt separately. The carbonyl functional group index was used to represent the carbonyl compounds in asphalt, and $I_{C=O}$ was used to characterize the content of carbonyl.

$$I_{Ar} = \frac{A_{1600\mathrm{cm}^{-1}}}{\sum A_{2000\mathrm{cm}^{-1}} - 600\mathrm{cm}^{-1}} \tag{1}$$

$$I_{B,a} = \frac{A_{1377\mathrm{cm}^{-1}}}{\sum A_{1377\mathrm{cm}^{-1}} + A_{1461\mathrm{cm}^{-1}}} \tag{2}$$

$$I_B = \frac{A_{2922\mathrm{cm}^{-1}} + A_{2850\mathrm{cm}^{-1}}}{\sum A_{2000\mathrm{cm}^{-1}} - 1600\mathrm{cm}^{-1}} \tag{3}$$

$$I_{c=C} = \frac{A_{966\mathrm{cm}^{-1}}}{\sum A_{2000\mathrm{cm}^{-1}} - 600\mathrm{cm}^{-1}} \tag{4}$$

$$I_{c-H} = \frac{A_{697\mathrm{cm}^{-1}}}{\sum A_{2000\mathrm{cm}^{-1}} - 600\mathrm{cm}^{-1}} \tag{5}$$

$$I_{c=o} = \frac{A_{1820\mathrm{cm}^{-1}} - 1680\mathrm{cm}^{-1}}{\sum A_{2000\mathrm{cm}^{-1}} - 600\mathrm{cm}^{-1}} \tag{6}$$

The infrared spectra were determined after the peak fitting process, and the infrared spectrum indexes are depicted in Figure 4.

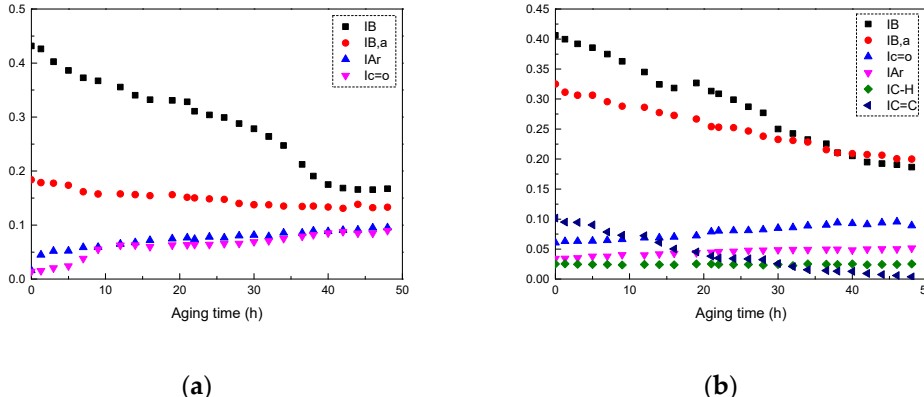

**Figure 4.** Infrared spectrum of asphalt: (**a**) base asphalt; (**b**) SBS-modified asphalt.

It can be seen from Figure 4 that the infrared spectrum indexes of base asphalt and SBS-modified asphalt show different trends with changes in thermal aging time.

Under the condition of thermal aging, the $I_{Ar}$ values and $I_{B,a}$ values of the base asphalt and SBS-modified asphalt increased and decreased with increasing aging time, respectively. When the aging time was 48 h, the $I_{Ar}$ value and $I_{B,a}$ value of base asphalt were about 7 times and 0.78 times those at the initial aging state, respectively, while the $I_{Ar}$ value and $I_{B,a}$ value of SBS-modified asphalt were about 2 times and 0.6 times those in the initial aging state, respectively. This shows that with the increase of aging time, the aging effect was gradually strengthened, and the content of aromatic functional groups and aromatic components gradually increased and decreased, respectively, resulting in the relative content of asphaltene and resins increasing. Additionally, by comparing the indexes of base asphalt and SBS-modified asphalt, it can be seen that in aromatic functional groups, the aromatic content of base asphalt decreased faster than that of SBS-modified asphalt, and the generation of resins and asphaltene were faster than in SBS-modified asphalt, that is, the aging resistance of the base asphalt was worse than that of SBS-modified asphalt.

With the increase in thermally aged time, the $I_B$ values of base asphalt and SBS-modified asphalt decreased. When the aging time was 1 h, the $I_B$ values of base asphalt and SBS-modified asphalt were about 2.3 times and 2.0 times those for aging time 48 h, respectively. That is, when the aging effect of asphalt was increased, the content of saturated components in asphalt decreased, and the decrease rate in base asphalt was faster than that of SBS-modified asphalt, indicating that the aging resistance of base asphalt was worse than that of SBS-modified asphalt.

Under different aging conditions, the $I_{C=C}$ value of SBS-modified asphalt decreased, while the $I_{C-H}$ value of SBS-modified asphalt was stable with increasing aging time. When the aging time was 48 h, the $I_{C=C}$ of SBS-modified asphalt was about 0.1 times its original value, and it can be concluded that thermal aging has a certain promotion effect on the cracking of butadiene polymer in SBS-modified asphalt, while having little effect on the cracking of polystyrene. Thus, the thermal aging of SBS-modified asphalt was mainly caused by the cracking of polymer butadiene.

Figure 4 shows that the carbonyl content of base asphalt and SBS-modified asphalt increased with the increase of aging time. When the aging time was 48 h, the $I_{C=O}$ values of base asphalt and SBS-modified asphalt were about 5 times and 1.3 times their initial state. The reason was that in the process of asphalt aging, oxygen reacts with polar groups in asphalt to produce carbonyl compounds, and with the increase in aging degree, and the stronger the reaction between oxygen and polar groups, the more carbonyl compounds are produced (as shown in Equation (7)). Therefore, the research on carbonyl functional group index is able to predict the aging state of asphalt. Additionally, from the $I_{C=O}$ growth rates

of base asphalt and SBS-modified asphalt, the aging resistance of base asphalt was worse than that of SBS-modified asphalt.

$$R-CH=CH_2+O_2 \xrightarrow[heating]{catalyst} R-CH_2- \overset{\overset{O}{\|}}{CH_2} \tag{7}$$

*3.3. Thermal Oxidative Aging Pattern Analysis of Asphalt Based on the Peak Fitting Method*

3.3.1. Two-Reaction Kinetic Model

Based on the Arrhenius kinetics model, a two-reaction kinetic aging model was developed by Jin [14].

The thermal aging process consists of two steps: the fast-rate reaction period and constant-rate reaction period, as expressed in Equations (8)–(11).

$$CA = CA_{\tan k} + M\left(1 - e^{-kft}\right) + k_c t \tag{8}$$

$$M = CA_0 - CA_{\tan k} \tag{9}$$

$$k_f = A_{fe}^{-\frac{Eaf}{RT}} \tag{10}$$

$$k_C = A_{Ce}^{-\frac{EaC}{RT}} \tag{11}$$

The above equations contain three factors. The first factor $CA_{\tan k}$ is the carbonyl area value of original asphalt; the second factor is the fast-rate reaction period of thermal aging, where $CA$ is the carbonyl area value of asphalt sample, $CA_0$ is the intercept of the constant line, $M$ is the difference between $CA_{\tan k}$ and $CA_0$, $t$(d) is the oxygen aging time of asphalt, and $K_f$(1/day) is the reaction constant of fast-rate reaction period; and the third factor is the constant-rate reaction period of thermal aging, where $K_c$(1/day) is the constant-rate reaction constant, $A_f$ and $A_c$ are pre-exponential factors of $K_f$ and $K_c$, $E_{af}$ and $E_{ac}$ are apparent activation energies of $K_f$ and $K_c$, $T$ is absolute temperature (436.15 K), and $R$ is ideal gas constant (8.314 J/mol/K).

3.3.2. Thermal Oxidative Aging Pattern Analysis

To study the thermally aged pattern of asphalt based on the peak fitting method, the mathematical method was performed to optimize and solve the parameters of the above two-reaction kinetic model. The six parameters ($CA_{\tan k}$, $M$, $A_f$, $A_c$, $E_{af}$, $E_{ac}$) in the thermal aging model of base asphalt and SBS-modified asphalt were simulated and optimized by the piecewise fitting method in OriginPro 9.0, in which the thermal aging time was $t$ an independent variable and $CA$ was a dependent variable. The fitting curves of the two kinds of asphalt are depicted in Figure 5, and the fitting parameters are shown in Table 4.

Figure 5 shows the thermally aged process of asphalt, and the curve slope shows the growth rate of the carbonyl content of asphalt at different aging times. It can be seen from the figure that the slope of the curve was higher in the early stage of aging, and the curve slope decreased gradually with increasing aging time. This change trend corresponded to the fast reaction period and constant-rate reaction period of thermally aged asphalt, and 12 h was about the intersection of the two periods. Through the comparison of M value in Table 5, it can be concluded that the growth rate of the carbonyl content of base asphalt was higher than SBS-modified asphalt. Therefore, it can be concluded from the model that the aging rate of base asphalt was higher than SBS-modified asphalt. In addition, according to Table 5, the thermally aged curve of asphalt based on the peak fitting method can be simulated by the piecewise fitting method of OriginPro 9.0, and the coefficient of determination was above 0.91. Therefore, the thermally aged process based on the peak fitting method conformed to the two-reaction kinetic model, and the carbonyl content of asphalt in the process of thermal aging can be predicted and evaluated accurately by the fitting curve and optimization parameters.

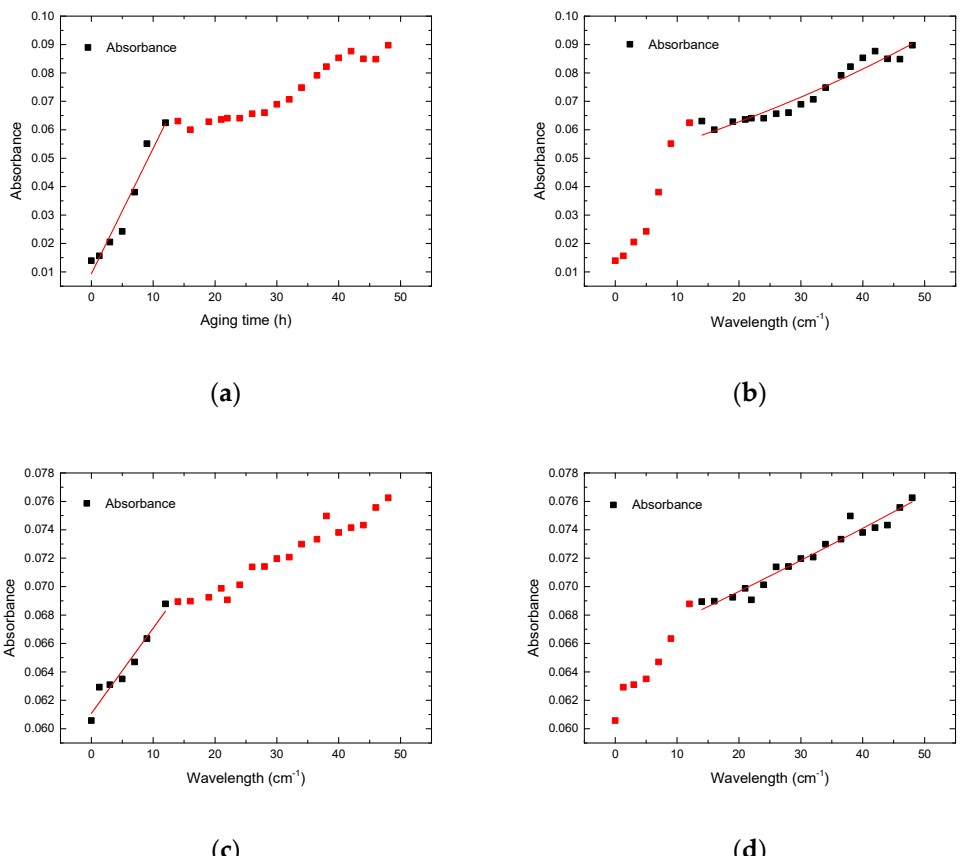

**Figure 5.** Thermal oxidative aging curve of base asphalt and SBS-modified asphalt based on peak fitting method: (**a**) fast-rate reaction period of base asphalt; (**b**) constant-rate reaction period of base asphalt; (**c**) fast-rate reaction period of SBS-modified asphalt; (**d**) constant-rate reaction period of SBS-modified asphalt.

**Table 5.** Optimization parameters of two kinds of thermally aged asphalt based on peak fitting method.

| Asphalt Type | Carbonyl Content of Original Asphalt | Fast-rate Reaction Period | | | | Constant-rate Reaction Period | | |
|---|---|---|---|---|---|---|---|---|
| | $CA_{tank}$ | $M$ | $A_f$ | $E_{af}$ | R2 | $A_c$ | $E_{ac}$ | R2 |
| Base asphalt | 0.0093 | 36.03 | $-1.22 \times 10^{-4}$ | 50.26 | 0.922 | 0.048 | $-46.959$ | 0.932 |
| SBS-modified asphalt | 0.0550 | 4.28 | $-1.42 \times 10^{-4}$ | 52.54 | 0.912 | 0.065 | $-11.205$ | 0.952 |

## 4. Conclusions

The thermally aged asphalt at different durations was tested and analyzed based on the peak fitting method of infrared spectrum. Additionally, the infrared spectrum group index of aging asphalt was measured, and the parameters of the thermally aged asphalt pattern were fitted based on the peak fitting method. The key findings of this work are:

(1) The infrared spectrum was used to qualitatively analyze the original asphalt. From the results, the base asphalt was composed of aliphatic compounds, aromatic compounds, and heteroatom derivatives. Additionally, the SBS-modified asphalt contained styrene butadiene styrene in SBS modifier compared to base asphalt.

(2) Based on the peak fitting method, the aromatic functional group index $I_{Ar}$, fat branched chain index $I_{B,a}$, aliphatic functional group index $I_B$, butadiene functional group index $I_{C=C}$, styrene index $I_{C-H}$ and carbonyl functional group index $I_{C=o}$

were proposed to estimate the thermally aged state of asphalt. With the increase in aging time, the aromatic functional group index $I_{Ar}$ and carbonyl group index $I_{C=o}$ increased, the fat branched chain index $I_{B,a}$, aliphatic group $I_B$ and butadiene functional group $I_{C=C}$ decreased, and the styrene index $I_{C-H}$ was stable. The results suggest that the carbonyl content of the aged asphalt was closely related to the aging degree of the asphalt, during the aging process of asphalt, the content of aromatic and saturated components decreased, while the content of gum and asphalt increased. Meanwhile, the aging effect can promote the cracking of polymer butadiene of SBS-modified asphalt to a certain extent, while the aging effect had little effect on the cracking of polystyrene; it was speculated that the thermally aged SBS-modified asphalt was mainly caused by the cracking of polymer butadiene.

(3) Taking the carbonyl functional group index as the research object, the thermally aged pattern of asphalt based on peak fitting method was evaluated. The aging pattern of thermally aging asphalt based on the peak fitting method conformed to the two-reaction kinetic model, and the aging process consists of a fast-rate reaction period and constant-rate reaction period. The parameters of the model were optimized by using the piecewise fitting method of OriginPro 9.0, and the coefficient of determinations are all above 0.91.

In summary, the peak fitting method can increase the calculation accuracy caused by overlapping and partial overlapping of infrared spectrum peaks, and the thermally aged pattern of asphalts based on peak fitting method were investigated. The results showed that the aging state of asphalts can be estimated by group indexes, the aging degree of asphalt can be predicted by the aging pattern based on the peak fitting method, and the peak fitting method conformed to the two-reaction kinetic model, in which the coefficient of determination can achieve 0.91.

**Author Contributions:** Formal analysis, Z.L. (Zhuilin Li), S.H. and K.Z.; Investigation, S.H.; Methodology, S.H.; Project administration, Z.L. (Zonghe Li); Resources, Z.L. (Zonghe Li); Software, Z.L. (Zhuolin Li) and P.Z.; Writing—review & editing, Z.L. (Zhuolin Li). All authors have read and agreed to the published version of the manuscript.

**Funding:** This work was supported by Research Project by the transportation department of Liaoning Province (Grant No. 201812), Natural Science Foundation of Liaoning Province (Grant No. 20170540498) and Key Laboratory of Transport Industry of Expressway Maintenance Technology. The authors gratefully acknowledge their financial support.

**Institutional Review Board Statement:** Not applicable.

**Informed Consent Statement:** Not applicable.

**Data Availability Statement:** Not applicable.

**Conflicts of Interest:** The authors declare no conflict of interest.

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
