# Peer review of "Aging Analysis of Thermally Aged Asphalt Using Peak-Fitting Method: Its Pattern and Statistical Prediction"

_coatings, doi:10.3390/coatings12050582_

Round 1

Reviewer 1 Report

Good, solid experimental work, well described and logically discussed. For example 15 runs of FTIR spectra were collected (line 93).

Just few minor editorial corrections:

line 66 change sentence ...parameters to be used in a pavement
oxidation
....  to sentence ...parameters to be used in an asphalt pavement
oxidation
... or sentence ...parameters to be used in an asphalt
oxidation
.....

line 88 - complete FTIR manufacturer, state (info in parenthesis)

line 99 - complete software version in sentence ...Peakfit soft
ware (version) was used....

line 210 - correct small k to capital K (two times)

line 338 - correct to ...International....

Author Response

Response to Reviewer 1 Comments

Dear review:

Thanks for your comments concerning our manuscript entitled “Aging analysis of thermally aged asphalt using peak-fitting method: its pattern and statistical prediction”(1694478). those comments are so precious and helpful for revising and improving our manuscript. We have studied the comments carefully and have made correction according to those comments. Revised portion are in red in the paper. The responds are as flows:

Point 1: line 66 change sentence ...parameters to be used in a pavement
oxidation.... to sentence ...parameters to be used in an asphalt pavement
oxidation... or sentence ...parameters to be used in an asphalt
oxidation.....

Response 1: The sentence has been revised as shown in line 71.

Point 2: line 88 - complete FTIR manufacturer, state (info in parenthesis)

Response 2: The information of FTIR has been revised as shown in line 94.

Point 3: line 99 - complete software version in sentence ...Peakfit soft
ware (version) was used....

Response 3: The version has been revised as shown in line 104.

Point 4: line 210 - correct small k to capital K (two times)

Response 4: The letters have been revised as shown in line 218.

Point 5: line 338 - correct to ...International....

Response 5: The word have been revised as shown in line347.

Reviewer 2 Report

The paper "Aging analysis of thermal aging asphalt using peak-fitting method: its pattern and statistical prediction" is within the scope of Coatings. I think the content is innovative and never published, but it should be improved. It studies the thermally aged pattern of asphalt based on the peak fitting method using statistical methods. Fourier transform infrared spectroscopy was used to investigate the modification of asphalt group due to the oxidative aging.

The paper should be improved:

  1. the abstract is only the summary of the study, but it does not justify the study and does not discuss the results and their effect on the state of the art;
  2. the introduction lists several studies, but they are not correctly presented. Often 4-8 references are listed at the end of a sentence.
  3. the first sentence of 2.1 Materials is not clear: Two distinct binders were considered in this study: base asphalt (Liaohe 90#) and SBS-modified asphalt (polymer occupies 5%). The study refers to binders (bitumen) or mixtures (asphalt)? according to Tables 1 and 2, the authors refer to bitumen, not to asphalt.
  4. the correct symbol for liter is l, not L
  5. what are the axis names in Figure 1?
  6. all the names and variables in Equations 1 to 6 should be explained;
  7. the legend in Figure 4 should be explained before the Figure is presented.

Author Response

Response to Reviewer 2 Comments

Dear review:

Thanks for your comments concerning our manuscript entitled “Aging analysis of thermally aged asphalt using peak-fitting method: its pattern and statistical prediction”(1694478). those comments are so precious and helpful for revising and improving our manuscript. We have studied the comments carefully and have made correction according to those comments. Revised portion are in red in the paper. The responds are as flows:

Point 1: The abstract is only the summary of the study, but it does not justify the study and does not discuss the results and their effect on the state of the art.

Response 1: The abstract has been revised as shown in line 24 to line 28.

Point 2: The introduction lists several studies, but they are not correctly presented. Often 4-8 references are listed at the end of a sentence.

Response 2: In this paper introduction, some research results belong to relevant research or further research. Therefore, several relevant literatures are listed as a whole at the end of a sentence, while for special results, literatures are listed separately in the introduction.

Point 3: The first sentence of 2.1 Materials is not clear: Two distinct binders were considered in this study: base asphalt (Liaohe 90#) and SBS-modified asphalt (polymer occupies 5%). The study refers to binders (bitumen) or mixtures (asphalt)? according to Tables 1 and 2, the authors refer to bitumen, not to asphalt.

Response 3: The word asphalt has been modified as bitumen as shown in line 76 to line 82.

Point 4: The correct symbol for liter is l, not L

Response 4: The symbol of liter has been revised as shown in line 87.

Point 5: What are the axis names in Figure 1?

Response 5:The Y-axis name is absorbance and the X-axis name is wavelength.

Point 6: All the names and variables in Equations 1 to 6 should be explained;

Response 6: The variables and the names have been introduced above the Equations.

Point 7: The legend in Figure 4 should be explained before the Figure is presented.

Response 7: The legend in Figure 4 has be explained before the Figure is presented.

Round 2

Reviewer 2 Report

The paper can be accepted

This manuscript is a resubmission of an earlier submission. The following is a list of the peer review reports and author responses from that submission.

Round 1

Reviewer 1 Report

The title needs to be revised to make it more pertinent and free of grammar error. (e.g., remove one “aging” word; revise “thermal aging asphalt” to “thermally aged asphalt”).

The literature review in the “Introduction” section is insufficient. What is the two-reaction kinetic model?

In the abstract and in the body text, R squared is known as coefficient of determination, not “fitting coefficient”.

In Section 2.4, the “peak fitting” method needs to be described instead of just showing a figure (Figure 1).  In the abstract, it is mentioned that it is a “piecewise fitting method”, but I do not see any indication of “piecewise” in Figure 1. The justification for using the Gaussian function in the peaking fitting method needs to described. How are the peaks selected during the model estimation process? Are different weights given to different Gaussian functions in the estimation procedure?

Line 95, why “R-squared was required to be above 0.90”?  An adjusted R-squared is more appropriate. The example shown in Figure 1 has an R-squared value of 0.91, but the plot shows some poor fitting at some peak frequencies (e.g., at the wavenumber of 1307.1). This suggest the peak fitting method may not a good method to characterize the FTIR results.

This manuscript emphasizes the use of so-called peak fitting method, but I do not see the necessity of using this method to analyze the FTIR results. I suggest the authors compare the results of the two-reaction kinetic model with and without the peak fitting method.

Line 73, “rolling thin film oven aging test” instead of “rotating film aging test”.

Line 130, the Lambert Beer law needs to be introduced.

Lines 142-156, something is missing in the phrases “the values and values of base asphalt” in several places.

In Lines 191 to 201, the notations in the equations are inconsistent with those in the text.

The reference items use different styles in the reference section, which should be consistent.

There are many grammar and format errors throughout the manuscript.  An extensive proofread is necessary in the revision of the manuscript. Some examples are as follows:

Line 33, “polar molecular”

Line 34, “which cause”

Line 40, “which of”

Line 54, “which overlook”

Lines 54-55, “which overlook.. are not…”

Line 56, “which will affects”

Line 60, spell out “CA”.

Line 69, “table1” and “table2”

Line 89, a period is missing

Reviewer 2 Report

Interesting paper describing practical application of OriginPro 9.0 software for advanced analysis and interpretation of FTIR spectra collected during aging of two asphalt samples. This is positive value of this paper. But the editorial form requires detailed correction. This is not complicated, majority of mistakes can be corrected with simple mouse clicks but this must be done. Paper in this form suggests lack of respect for the reader what definitely was not an idea of the Authors. Some of these mistakes are listed below:

page 2 line 46 - there ought to be space between digits and units (except % and 0C) - so it should be ... 1650 cm-1... 1820 cm-1...

page 2 line 49 - should be ...carbonyl...

page 2 line 60 - waht is meaning CA ? Please explain.

page 2 line 69 - should be ...Table 1 and Table 2...

page 2 line 73 - check font size of this sentence

page 2 line 76 - correct this sentence, this is not an air pressure, this is an air flow rate, it was probably done at the atmospheric pressure

page 3 Table 3 - should be ...temperature (K)...

page 3 lines 82 and 83 - check font size ot these sentences

page 4 lines 106; 108; 109; 110; 118 - add space (see remark number one above)

page 4 Table 3 - add space two times in ...Calcite (carbonate....)

page 5 lines 131; 132 - dd space (see remark number one above) 

page 6 Figure 4 - add space ...(a) base....

page 6 - always add space after numer, for example it should be in line 142: ...(1) The aromatic...

page 6 line 170 - something is missing in this sentence

page 7 formula (7) - should be ...catalyst...

page 7 line 201 - should be...K...

page 7 line 210 - the Table number should be 4 no 3

page 8 line 215; 223 - the same remark as above

page 8 line 229 - should be...model, and ...

page 9 - the same remark as page 6 line 142

Please make a comment how were changing during aging the mechanical properties of tested two asphalts.

Please check also Referencies (14; 29; 33; 34; 35; 37)

Round 2

Reviewer 1 Report

I have no further comments on the technical aspect of the manuscript. There are still grammar errors in it, so proofreading is needed before it can be considered for publication.